# Strength Characteristics of Controlled Low-Strength Materials with Waste Paper Sludge Ash (WPSA) for Prevention of Sewage Pipe Damage

**DOI:** 10.3390/ma13194238

**Published:** 2020-09-23

**Authors:** Jeongjun Park, Gigwon Hong

**Affiliations:** 1Incheon Disaster Prevention Research Center, Incheon National University, Incheon 22012, Korea; smearjun@hanmail.net; 2Institute of Technology Research and Development, Korea Engineering & Construction, Seoul 05661, Korea

**Keywords:** waste paper sludge ash (WPSA), controlled low-strength material (CLSM), unconfined compressive strength, bearing capacity, backfill material

## Abstract

In this study, the effects of the mixing conditions of waste paper sludge ash (WPSA) on the strength and bearing capacity of controlled low-strength material (CLSM) were evaluated, and the optimal mixing conditions were used to evaluate the strength characteristics of CLSM with recyclable WPSA. The strength and bearing capacity of CLSM with WPSA were evaluated using unconfined compressive strength tests and plate bearing tests, respectively. The unconfined compressive strength test results show that the optimal mixing conditions for securing 0.8–1.2 MPa of target strength under 5% of cement content conditions can be obtained when both WPSA and fly ash are used. This is because WPSA and fly ash, which act as binders, have a significant impact on overall strength when the cement content is low. The bearing capacity of weathered soil increased from 550 to 575 kPa over time, and CLSM with WPSA increased significantly, from 560 to 730 kPa. This means that the bearing capacity of CLSM with WPSA was 2.0% higher than that of weathered soil immediately after construction; furthermore, it was 27% higher at 60 days of age. In addition, the allowable bearing capacity of CLSM corresponding to the optimal mixing conditions was evaluated, and it was found that this value increased by 30.4% until 60 days of age. This increase rate was 6.7 times larger than that of weathered soil (4.5%). Therefore, based on the allowable bearing capacity calculation results, CLSM with WPSA was applied as a sewage pipe backfill material. It was found that CLSM with WPSA performed better as backfill and was more stable than soil immediately after construction. The results of this study confirm that CLSM with WPSA can be utilized as sewage pipe backfill material.

## 1. Introduction

With the aging of buried pipes, incidents of ground subsidence around them have become increasingly frequent in South Korea. Underground cavities around sewage pipes in urban areas caused by aging and damage have been found to be the main cause of ground subsidence, accounting for approximately 82% of the total number of occurrences, as shown in Figure 1 [1,2].

When aged sewage pipes are replaced, the ground is excavated and soil is backfilled after new sewage pipes are placed. The soil excavated on site is used as backfill in many cases, but there are also many cases in which the compaction of the ground around the pipes is not sufficient. The reason is that the compaction energy by equipment may affect the damage to sewer pipes when such compaction is performed. This problem can occur when the hydraulic filling of high-quality sand is difficult at the construction site. Specifically, it is a general restoration method that must be restored immediately after excavation construction, in order to secure the usability of the surface ground over sewage pipes, such as roads.

Poor compaction of the ground around sewage pipes causes stress concentration on the pipes when loaded from above, causing long-term damage as the pipes age. As this causes underground cavities that lead to ground subsidence around sewage pipes, it is necessary to use backfill materials that provide sufficient bearing capacity and stability of the ground around the pipes.

Studies on various backfill materials and methods have been conducted to determine how to prevent sewage pipe damage and ensure sufficient bearing capacity of the surrounding ground [3]. Controlled low-strength material (CLSM) was developed to address various problems caused by inadequate backfill materials and poor compaction [4].

CLSM is a construction material that applies the concept of low-strength concrete to geotechnical engineering. CLSM, which was developed in the United States in the mid-1900s, has been used as a backfill material for buried pipes, such as water and sewage pipes. The American Concrete Institute (ACI) has defined CLSM as material composed of sand, cement, admixture, fly ash, and water, and has suggested that the strength range be determined according to the method of excavation. For example, the suggested unconfined compressive strength is less than approximately 0.3 MPa when excavation is performed using manpower, 0.7–1.4 MPa when excavation is performed using machinery, and less than 8.3 MPa when excavation is not required. Unlike conventional soil backfill materials, CLSM is self-leveling, self-compacting, and flowable. In addition, it is possible to control its strength. Therefore, it can be used as a countermeasure against ground subsidence caused by the faulty installation of buried pipes and soil loss underground, as shown in Figure 2. It is also highly useful as an eco-friendly material because various recycled materials can be incorporated into it [5].

In recent years, various studies have been conducted on CLSM with coal ash as the main component to explore the feasibility of utilizing coal ash in CLSM on a large scale. Kim et al. [7] conducted unconfined compressive strength tests of CLSM with coal ash as the main component, in an effort to develop various construction materials and analyzed strength as a function of curing time and water content. Kong et al. [8] evaluated the strength characteristics of CLSM composed of fly ash, cement, water, and pond ash in place of sand, depending on the mixing ratio between pond ash and fly ash. Won and Lee [9] conducted research on the basic properties and strength characteristics of a CLSM mixture with bottom ash and a mixture with fly ash. Razak et al. [10] prepared a CLSM mixture using bottom ash and evaluated its performance under various curing conditions.

While several studies have been conducted on CLSM using coal ash as the main component, several others have been conducted on the development of new concrete mixtures using materials such as recycled concrete aggregate (RCA) and supplementary cementitious material (SCM) as replacements for Portland cement, a component of CLSM.

Hao et al. [11] evaluated the compressive and tensile strength of CLSM mixtures with fly ash, bottom ash, and paper sludge used as substitutes for cement and conducted research on methods for reducing carbon dioxide (CO_2_) emissions. Kubissa et al. [12] measured compressive strength in the range of 0.52–4.29 MPa for CLSM containing fly ash and RCA. They also developed a mixture that achieved the same performance as conventional CLSM, using only a small amount of cement. 

Naganathan et al. [13] prepared a CLSM mixture in which the amount of cement was minimized using RCA and fly ash, and they confirmed that the addition of fly ash improved the strength of the mixture. Ahmadi and Al-Khaja [14] compared the chemical properties of concrete mixtures with various paper sludge mixing ratios with those of existing concrete mixtures and confirmed that paper sludge can be used as a construction material. Frías et al. [15] and García et al. [16] developed a material that could be transformed into metakaolinite through the calcination of paper sludge, and used the material to evaluate the applicability of paper sludge as a cement substitute and its environmental impact through recycling. Monzó et al. [17], Horiguchi et al. [18], and Boni et al. [19] conducted research on the feasibility of replacing Portland cement with paper sludge, sewage sludge, and the waste obtained by incinerating them and derived management plans for industrial waste from an environmental perspective.

The amount of waste generated is increasing exponentially every year, due to the growing consumption caused by the growth of cities, and this increasing generation of waste is causing serious environmental problems. Therefore, it is urgent to identify ways to treat the ash emitted in large quantities by the incineration of waste and develop treatment technologies for such waste [20]. 

Fly ash and waste paper sludge ash (WPSA), which is obtained by incinerating paper sludge, are typical products of waste incineration. Various studies have been conducted on recycling WPSA. Heo et al. [21] reported that WPSA can be used as a lightweight embankment material because it has excellent engineering properties in comparison to fly ash and soil. They also noted that WPSA causes no problems in terms of environmental impact, because the proportion of lime (CaO) is nine times higher than that of fly ash and the concentrations of toxic heavy metals are lower than the established thresholds. 

Lee et al. [22] conducted research on recycling WPSA as a construction material, by analyzing the quality characteristics of concrete and clay bricks fabricated by mixing with WPSA. Seo [23] evaluated the physical and engineering properties achieved when WPSA and fly ash were used as cement mixing materials. The possibility of stabilizing soil by improving the strength of clay using WPSA was evaluated in [24]. Bujulu et al. [25] evaluated the feasibility of using WPSA as a replacement for lime and cement in the stabilization of quick-clay and found that up to 50% of the lime and cement normally used can be replaced with WPSA. 

Ahmad et al. [26] confirmed that WPSA is suitable for use as a concrete substitute by comparing the compressive and tensile strength of concrete, in which 5–20% cement was replaced with WPSA with those of ordinary concrete. Sani et al. [27] replaced ordinary Portland cement with 50–100% WPSA and compared the compressive strength obtained as a function of curing time. Ridzuan et al. [28] investigated the mixing characteristics of RCA and WPSA when used to replace Portland cement and evaluated the strength of CLSM based on the optimal content of WPSA (which yielded the maximum strength). Fauzi et al. [29] and Azmi et al. [30] evaluated the strength of WPSA and RCA mixtures. Bai et al. [31] and Mozaffari et al. [32] evaluated the applicability of WPSA and blast furnace slag through physical and chemical evaluation, using them as mixing materials and comparing the strength of mixes containing them with mixes containing ordinary Portland cement. 

Concrete and grout materials such as CLSM have an important correlation with the environmental characteristics and material improvement belowground. Farzampour [33,34] analyzed the strength of concrete and grout materials according to various temperature and humidity conditions. The results confirmed that the behavior of concrete and grout material has a greater effect on temperature than humidity. In addition, it was found that the material does not reach its ultimate strength when the temperature is significantly low in the curing conditions. Mansouri et al. [35] analyzed the structural behavior of concrete, considering the steel fiber effect and curing time in order to improve its abrasion resistance. It was confirmed that abrasion resistance increased when the steel fiber ratio was high.

The results of various studies confirm that WPSA can be used to address various problems associated with waste treatment, the unavailability of natural aggregates, and environmental concerns. Studies have shown that CLSM with WPSA can be used as a backfill material to prevent sewage pipe damage, by ensuring sufficient bearing capacity. In this study, the strength and bearing capacity characteristics of CLSM with WPSA were analyzed. The strength characteristics were evaluated by conducting unconfined compressive strength tests of mixtures with various WPSA mixing ratios, and the bearing capacity was evaluated by conducting plate bearing tests.

## 2. Strength Evaluation of CLSM with WPSA 

To evaluate the strength characteristics of CLSM with WPSA, a mix design process for determining the proportions of components (cement, sand, fly ash, and WPSA) was performed. In addition, flowability tests and unconfined compressive strength tests were conducted.

### 2.1. Materials

The CLSM used in this study was prepared by adding WPSA to conventional CLSM materials, resulting in mixtures of cement, sand, fly ash, and WPSA. Ordinary Portland cement was used as the cement, and Jumunjin standard sand was used as the sand. Table 1 lists the engineering properties of the standard sand. 

Fly ash was obtained from Ekons Co., Ltd. (Incheon, South Korea), and its main components were analyzed. The specific gravity of fly ash is 2.3, which is two-thirds that of ordinary cement, and it is known that the value increases as the iron content increases [36]. As Table 2 shows, SiO_2_ and Al_2_O_3_ made up more than 90% of the fly ash used in this study, indicating that reactive oxides that can be used in the polymerization reaction were present in large quantities. SiO_2_ can improve compressive strength in the long term, because it generates calcium silicate when it reacts with Ca(OH)_2_, which is generated when cement undergoes hydration.

Scanning electron microscopy (SEM) imaging was performed using equipment (model; JSM-7001F) manufactured by JEOL Ltd., Tokyo, Japan, to analyze the structural characteristics of the fly ash. The results are shown in Figure 3. The particle size of the fly ash ranged from 1 to 100 μm, with an average particle diameter in the range of 20 to 30 μm, which is very similar to the particle size characteristics of cement. The fly ash particles were smooth spheres, with pores observed on the surfaces of relatively large particles. XRD (X-ray diffraction) analysis of fly ash was performed using equipment (model; smartlab) manufactured by Rigaku, Tokyo, Japan, and it showed that quartz (SiO_2_) and mullite (3Al_2_O_3_·2SiO_2_) were present as crystalline substances, as shown in Figure 4. The general principle and operation of SEM and XRD can be confirmed in the research of Joseph et al. [37] and Borchert [38], respectively.

The WPSA used in this study was obtained from Ekons Co., Ltd. (Incheon, South Korea). The results of the analysis of its main components are as follows. The specific gravity was 2.5, and CaO, SiO_2_, and Al_2_O_3_ accounted for more than 80% of the main components (Table 3). CaO, which is a main component of cement, contributes to the excellent strength development of cement. Therefore, it is possible to adjust the strength development by adjusting the WPSA content. SEM imaging results for the WPSA showed that it was composed of ash particles and unburned fibers, with particle size ranging from to 2 to 100 μm. The WPSA included both spherical particles with smooth surfaces and plate-shaped particles (Figure 5). The results of XRD analysis of the WPSA confirmed that it was composed of CaO, CaCO_3_, and C_12_A_7_, as shown in Figure 6.

### 2.2. Mix Design of CLSM

As the CLSM used in this study was intended for use as backfill material for sewage pipes, re-excavation in the future must be easy, and the time required for construction needs to be minimized. As mentioned previously, ACI recommends an unconfined compressive strength in the range of 0.7–1.4 MPa for CLSM when mechanical excavation is conducted. In this study, the criterion for unconfined compressive strength at an age of 28 days was determined to be 0.8–1.2 MPa, based on previous studies, because the CLSM developed in this study must allow re-excavation for the maintenance of sewage pipes.

Mix designs with cement proportions of 5% and 10% and sand proportions of 35, 40, 45, and 50 were developed to assess the effects of the proportions of cement and sand on the strength characteristics of CLSM. For each combination of cement and sand, mix designs with WPSA-to-fly-ash ratios of 1:0, 1:1, and 0:1 were developed. The mix design was determined for a relative quantitative comparison of WPSA and FA, considering the ratio of cement and sand. Table 4 summarizes the mix designs.

### 2.3. Experiment Details

ACI specifies that materials with excellent flowability should not exhibit noticeable material segregation, and their flowability value must be at least 200 mm. Flowability tests were conducted in accordance with ASTM (American Society for Testing and Materials) [39] to ensure water content conditions that would satisfy the flowability requirements for each mix design. 

Unconfined compressive strength tests were conducted in accordance with ASTM [40]. In general, the unconfined compressive strength of a concrete material is based on a curing time of 28 days, but there are cases in which earlier strength information is required, depending on the use of the material. CLSM, which was used in this study, must perform adequately immediately after construction, because it is used as backfill material for sewage pipes. Emery and Johnston [41] proposed a value of 0.1 MPa for the strength of CLSM at 1 day of age. NRMCA (National Ready Mixed Concrete Association) [42] and Crouch et al. [43] proposed a range of 0.1–0.5 MPa for the strength of CLSM at 3 days of age. In this study, the criteria set for strength at 1 and 28 days of age were 0.1 MPa and 0.8–1.2 MPa, respectively. Unconfined compressive strength was measured at curing times of 1, 7, 28, and 60 days, the latter to assess the long-term strength characteristics of CLSM.

The CLSM specimens used in the unconfined compression strength tests were 100 mm (D) × 200 mm (H) and were fabricated by producing a mix with appropriate water content for each mix design, based on the flowability test results. After initial curing, the specimens were subjected to water curing. Three specimens were tested for each curing time, and the average value was calculated. The specimens were subjected to unconfined compression at a rate of 1 mm/min. Photographs in Figure 7 illustrate the test procedure.

### 2.4. Flowability and Unconfined Compressive Strength

Figure 8 and Table 5 show the flowability test results for the various mix designs. The water content that satisfied the flowability criterion (200 mm) ranged from 24 to 32%. When the proportions of WPSA and fly ash were identical, the water content required to ensure flowability decreased as the amount of sand increased. In addition, the required water content increased as the WPSA content increased.

Table 6 presents the results of the unconfined compressive strength tests for the various mix designs, showing the average value of the results of three specimens, with minimal deviation for each curing time. The unconfined compressive strength tended to increase as the curing time increased, regardless of the mix design, and increased very little if at all after 28 days of age. The unconfined compressive strength increased as the proportion of cement increased, all other aspects of the mix design being equal. Based on these test results, the effects of various aspects of the mix designs on the unconfined compressive strength characteristics were analyzed.

## 3. Results and Discussion

### 3.1. Unconfined Compressive Strength Versus Mixing Ratio between WPSA and Fly Ash 

For cases in which only WPSA was used (i.e., WPSA/FA ratio of 1:0), the effect of the WPSA content on the unconfined compressive strength was assessed, as shown in Figure 9a,b. Regardless of the cement content (5% or 10%), the unconfined compressive strength increased steadily from 1 to 28 days of age, and no significant strength change occurred after 28 days. The strength increased as the WPSA content increased, and was thus lowest when the WPSA content was lowest (cases 10 and 22). When the cement content was high (10%), the strength was higher than when the cement content was low (5%), because the WPSA content was relatively lower.

When the mixing ratio between WPSA and fly ash was 1:1, the unconfined compressive strength of CLSM increased steadily from 1 to 28 days of age for all mix designs, but did not increase significantly beyond 28 days, as shown in Figure 9c,d. When the cement content was 5%, the strength decreased as the sand content increased. However, when the cement content was 10%, the strength characteristics differed depending on the sand content. This may be because both WPSA and fly ash, which act as binders, have a significant impact on the overall strength when the cement content is low (5%), whereas sand has a larger impact on the strength characteristics than ash materials when the cement content is high (10%). 

Figure 9e,f show the unconfined compressive strength of CLSM containing only fly ash (i.e., WPSA/FA ratio of 0:1). When the cement content was 5%, the mix with the highest fly ash content exhibited the highest unconfined compressive strength. However, when the cement content was 10%, the mix with the lowest fly ash content exhibited the highest unconfined compressive strength. The strength increased significantly as the cement content increased, all other aspects of the mix design being equal. Especially, the result of case 21 shows that FA, cement, and sand are the mixing conditions with the maximum unconfined compressive strength when only FA is applied to CLSM. These results indicate that fly ash has a significant influence on the strength development of CLSM. 

### 3.2. Unconfined Compressive Strength Versus Sand Content 

The effect of the cement content on the unconfined compressive strength for a given sand content was evaluated. As shown in Figure 10, the rates of strength increase and strength achieved were significantly higher when the cement content was higher, regardless of the sand content and WPSA/fly ash mixing ratio. The lowest rate of increase in unconfined compressive strength was 18.3% when the sand content was 45%, and the highest was 593% when the sand content was 50%. As mentioned above, it was confirmed that, for the range of sand content considered, the rate of strength increase was highest when only fly ash was used. 

Mix designs that minimized the cement and sand content were selected to evaluate the applicability of CLSM by recycling WPSA. For cement content of 5%, cases 1, 2, and 8 were found to yield appropriate mix characteristics. For cement content of 10%, cases 16 and 19 were found to yield appropriate mix characteristics. In this study, however, the strength criteria for 28 days (0.8–1.2 MPa) and 1 day (0.1 MPa) of age were determined based on findings from previous studies on how to ensure the desired excavation conditions. As such, case 8, which corresponded to stable strength characteristics, was determined to be the optimal mix design, based on short-term (1 day) and long-term (60 day) strength, as well as cement content.

## 4. Bearing Capacity Evaluation of CLSM Using Plate Bearing Test

A site was prepared for plate bearing tests of CLSM with WPSA. Plate bearing tests were conducted at a location where CLSM was produced according to the mix design for case 8, which was determined to be the optimal mix design, and was used as backfill material at a location where weathered granite soil, which is commonly used as backfill material, was used. Based on the test results, the bearing capacity of the CLSM backfill and weathered granite soil backfill was evaluated.

### 4.1. Materials

Various laboratory tests were conducted to assess the engineering properties of the weathered soil used as control backfill. The specific gravity was 2.66 and the USCS (Unified soil classification system) soil classification was SP (Figure 11a). A maximum dry unit weight of 20.2 kN/m^3^ was observed at an optimal water content of 11.7% (Figure 11b).

The cohesion and internal friction angle of the weathered soil determined from direct shear testing were compared with those of CLSM. For this purpose, the CLSM test results at 7 and 28 days of age, when the rate of increase of unconfined compressive strength was high, were used. As shown in Figure 11c and Table 7, CLSM exhibited significantly higher shear strength than weathered soil. The cohesion and internal friction angle of CLSM after 28 days were 368% and 23% higher, respectively, than those of weathered soil.

### 4.2. Plate Bearing Test Procedure

At the site, a large soil tank was installed in the ground. The tank was separated in the middle so that the plate bearing test could be conducted, depending on the application of CLSM. The load was applied in six steps, in each of which the load was less than 98 kPa, or one-sixth of the test target load. After the load was increased in each loading step, it was maintained for at least 15 min. Ground settlement was measured at 1, 2, 3, 5, 10, and 15 min from the time of loading to 15 min, while the load was held constant using an LVDT (Linear Variable Differential Transformer) with an accuracy of 0.01 mm. It was assumed that settlement had stopped when settlement was less than 0.01 mm after 15 min or less than 1% of cumulative settlement after 1 min. Plate bearing tests were conducted at 1, 7, 14, 28, and 60 days of age to assess the effect of curing time on unconfined compressive strength, and the yield strength was expressed by a P–S curve developed from the test results. Figure 12 shows photographs of the plate bearing test procedure.

### 4.3. Evaluation Results

Figure 13 shows the plate bearing test results for the ground backfilled with weathered soil, for comparison with the bearing capacity of the ground where CLSM was applied. The P–S curve was used to calculate the load, corresponding to 10% settlement of the load plate diameter. A safety factor of 3 was applied to the allowable bearing capacity, based on the consideration that continuous cyclic loading could be applied to the ground in which sewage pipes were buried. It was found that the load corresponding to settlement equal to 10% of the load plate diameter increased from 550 to 575 kPa as the curing time increased. The bearing capacity of the weathered soil appeared to increase, because ground compaction was achieved by the tests conducted on selected curing days, but the rate of increase was judged to be insignificant.

Figure 14 shows the plate bearing test results for the ground where CLSM with WPSA was applied. When the results were analyzed in the same manner as those for weathered soil, it was found that the load increased from 560 to 730 kPa, as the curing time increased. When the rate of load increase was evaluated with respect to curing time, the load was found to increase by 20.5%, 24.1%, 28.6%, and 30.4% at 7, 14, 28, and 60 days of age, relative to the load at 1 day. These results show that the load increased continuously up to 28 days, but the rate of increase was not high after that. This trend is similar to that observed in the unconfined compression test results.

Figure 15 and Table 8 show the allowable bearing capacity of weathered soil and CLSM calculated from the plate bearing test results. As mentioned previously, the allowable bearing capacity was calculated by applying a safety factor of 3 to the load measured in the plate bearing test. The test results for weathered soil show that the allowable bearing capacity increased from 1 to 60 days of age by approximately 4%. In the case of ground where CLSM was applied, however, the allowable bearing capacity gradually increased as the curing time increased; it increased most significantly from 1 to 7 days of age, and then slowly after 28 days. The weathered soil and CLSM exhibited similar allowable bearing capacity immediately after construction.

These results suggest that when CLSM with WPSA is used as sewage pipe backfill material, it is possible to achieve acceptable performance immediately after construction. In addition, it was confirmed that CLSM with WPSA can be used as a backfill material that ensures higher stability than conventional soil backfill, because its allowable bearing capacity increases over time.

## 5. Conclusions

In this study, the strength characteristics of controlled low-strength material (CLSM) made with recyclable waste paper sludge ash (WPSA) were evaluated, to assess its ability to prevent ground subsidence caused by poor compaction of the ground around sewage pipes and ensure adequate bearing capacity of the ground above the pipes. The unconfined compressive strength of WPSA was evaluated with respect to the mixing conditions, and the bearing capacity of CLSM produced with the optimal mixing conditions was evaluated. The results of this study are summarized as follows: 

(1) When only WPSA was used, unconfined compressive strength increased steadily over time from 1 to 28 days, regardless of the cement content (5% or 10%). The strength did not change significantly after 28 days. Unconfined compressive strength increased as WPSA content increased.

(2) When the mixing ratio of WPSA to fly ash was 1:1, both materials, which act as binders, had a significant impact on strength when the cement content was low, but sand had a larger impact on the strength characteristics than ash material when the cement content was high.

(3) The strength and rate of strength increase were significantly higher when the cement content was high than when it was low, regardless of the sand content and WPSA–fly ash mixing ratio. The rate of strength increase was highest when only fly ash was used. 

(4) The strength corresponding to 10% settlement of the loading plate diameter was calculated based on the load–settlement relationship of the plate bearing test results. The strength of weathered soil increased from 550 to 575 kPa with increased age, and the increase rate of strength decreased with increased age. Furthermore, CLSM with WPSA increased from 560 to 730 kPa. In addition, the increase rate of CLSM strength increased, and then decreased after 28 days of age; this tendency was the same in weathered soil.

(5) The allowable bearing capacity of weathered soil and CLSM was calculated from plate bearing test results. The allowable bearing capacity of weathered soil increased by approximately 4.5%, with aging from 1 to 60 days. In the case of the ground where CLSM was applied, the allowable bearing capacity gradually increased as the curing time increased. Weathered soil and CLSM exhibited similar allowable bearing capacity immediately after construction. 

The results of this study confirm that CLSM with WPSA can be utilized as a sewage pipe backfill material, that can ensure higher stability than soil backfill. However, limited mixing conditions of WPSA and FA were applied in this study. Therefore, it is necessary to perform experiments and analyses on subdivided mixing conditions of WPSA and FA, in order to analyze the effect of WPSA on the strength of CLSM in detail.

## Figures and Tables

**Figure 1 materials-13-04238-f001:**
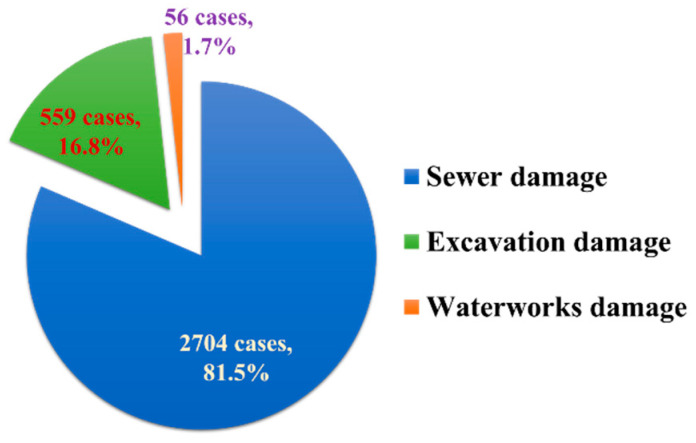
Causes of ground subsidence [1].

**Figure 2 materials-13-04238-f002:**
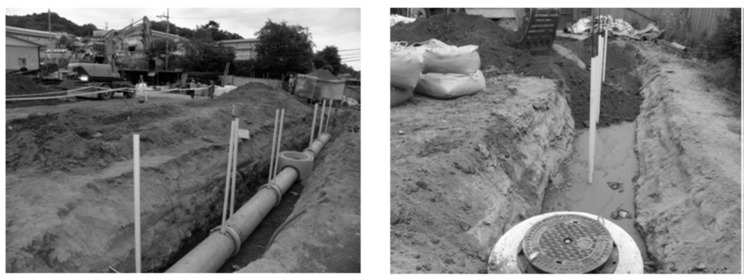
Backfilling of buried pipes using controlled low-strength material (CLSM) [6].

**Figure 3 materials-13-04238-f003:**
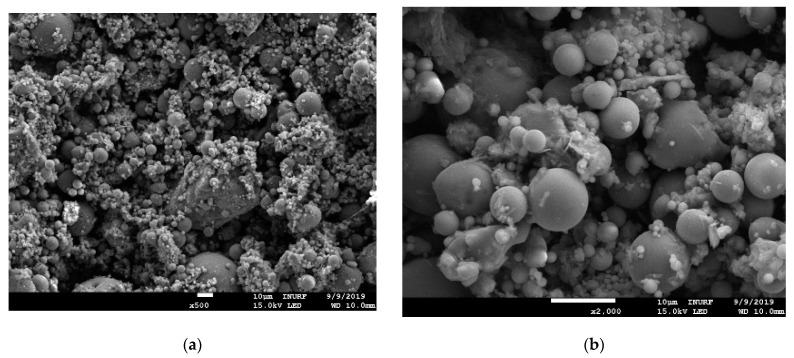
SEM images of fly ash: (**a**) 500×; (**b**) 2000×.

**Figure 4 materials-13-04238-f004:**
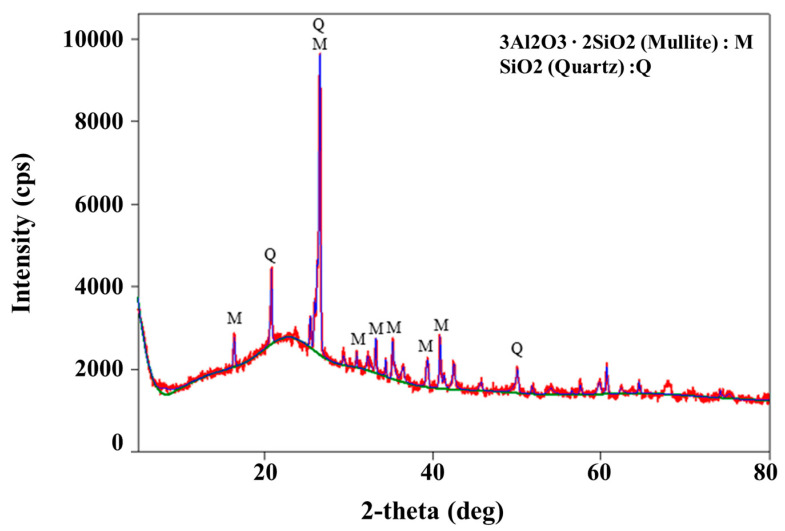
XRD results for fly ash.

**Figure 5 materials-13-04238-f005:**
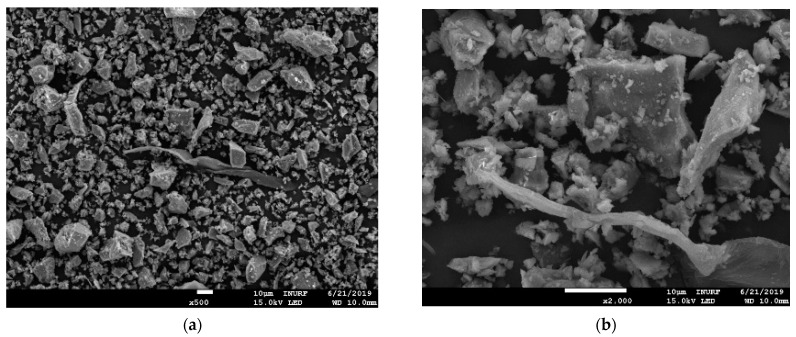
SEM images of WPSA: (**a**) 500×; (**b**) 2000×.

**Figure 6 materials-13-04238-f006:**
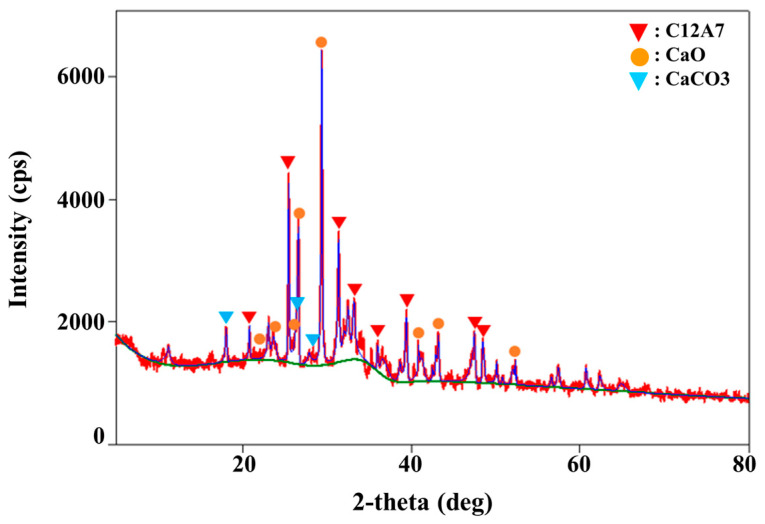
XRD results for WPSA.

**Figure 7 materials-13-04238-f007:**
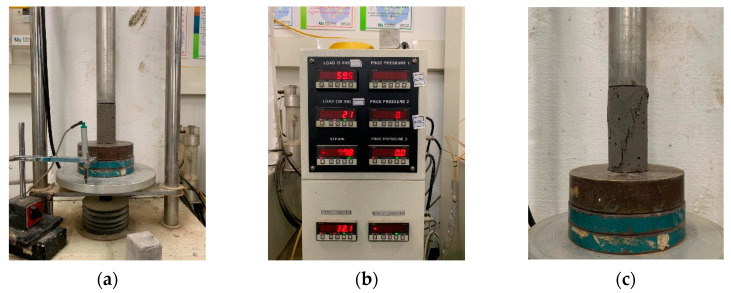
Procedure for unconfined compressive strength test: (**a**) specimen placement; (**b**) measurement of unconfined compressive strength; (**c**) specimen failure.

**Figure 8 materials-13-04238-f008:**
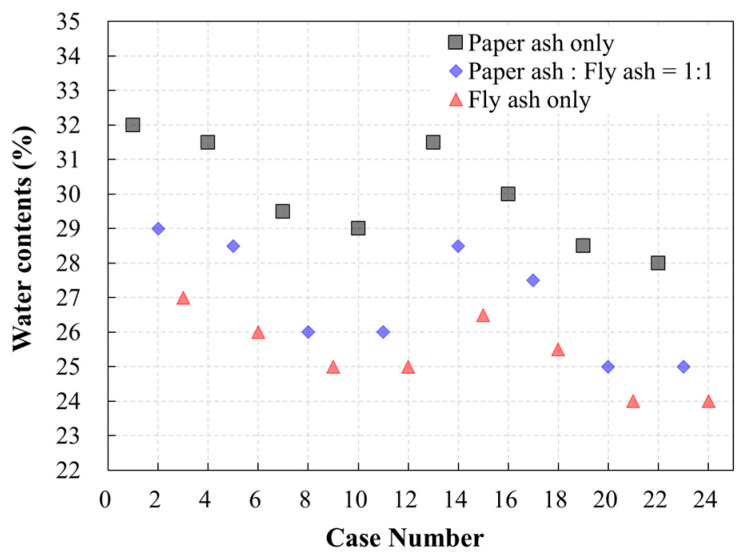
Water content required for adequate flowability.

**Figure 9 materials-13-04238-f009:**
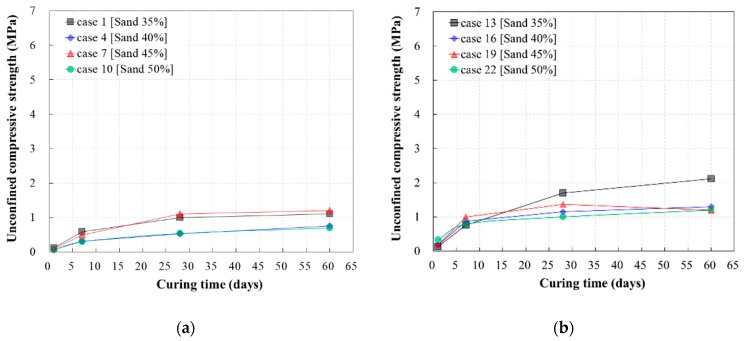
Unconfined compressive strength versus mixing ratio between WPSA and fly ash: (**a**) WPSA/FA = 1:0 (cement 5%); (**b**) WPSA/FA = 1:0 (cement 10%); (**c**) WPSA/FA = 1:1 (cement 5%); (**d**) WPSA/FA = 1:1 (cement 10%); (**e**) WPSA/FA = 0:1 (cement 5%); (**f**) WPSA/FA = 0:1 (cement 10%).

**Figure 10 materials-13-04238-f010:**
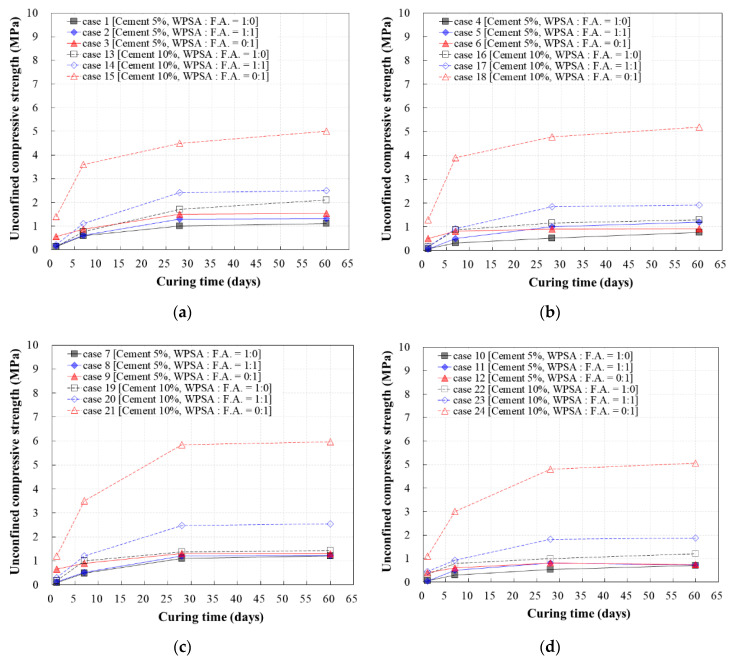
Unconfined compressive strength versus sand content: (**a**) sand 35%; (**b**) sand 40%; (**c**) sand 45%; (**d**) sand 50%.

**Figure 11 materials-13-04238-f011:**
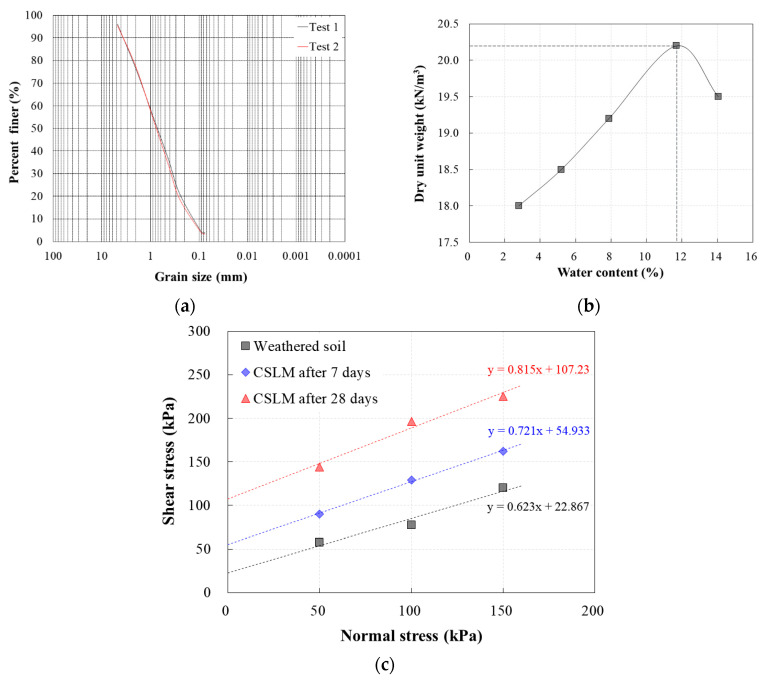
Engineering properties of weathered soil and CLSM: results of (**a**) sieve analysis of weathered soil; (**b**) compaction tests of weathered soil; (**c**) direct shear test.

**Figure 12 materials-13-04238-f012:**
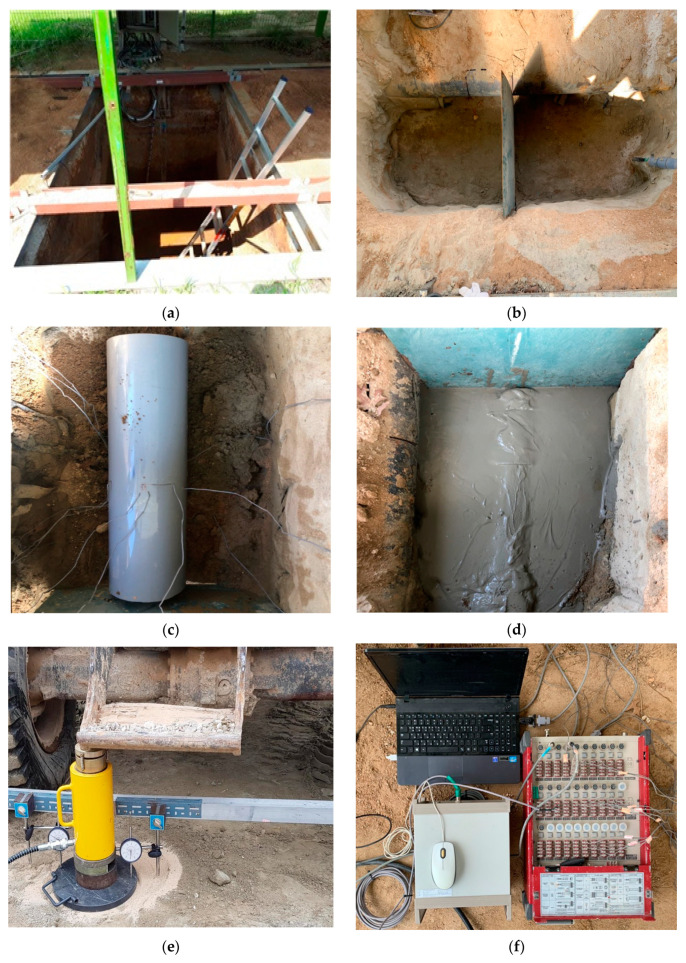
Plate bearing test procedure: (**a**) ground excavation and soil box installation; (**b**) ground construction; (**c**) buried sewage pipe; (**d**) CLSM construction; (**e**) plate bearing test; (**f**) test measurement.

**Figure 13 materials-13-04238-f013:**
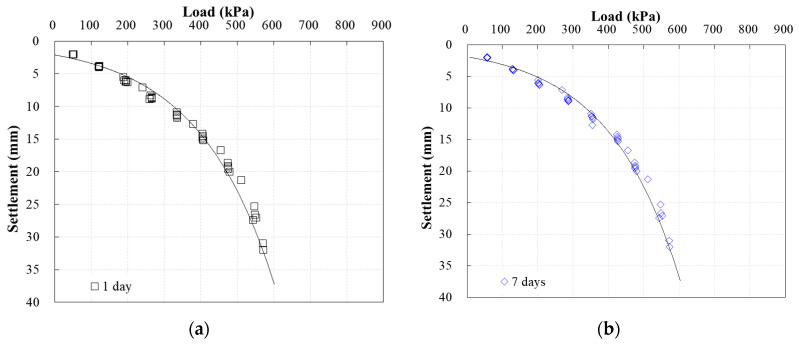
Plate bearing test results for weathered soil over time, with elapsed time of (**a**) 1 day; (**b**) 7 days; (**c**) 14 days; (**d**) 28 days; (**e**) 60 days.

**Figure 14 materials-13-04238-f014:**
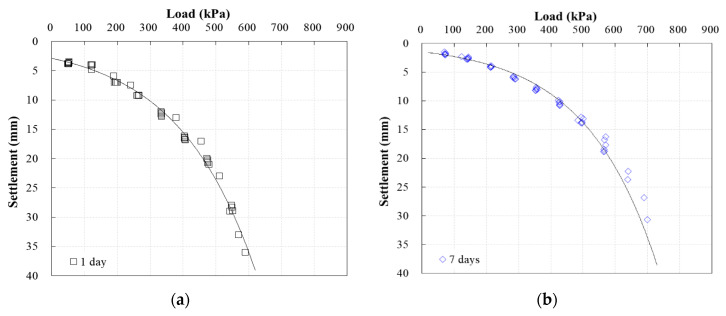
Plate bearing test results for CLSM over time, with elapsed time of (**a**) 1 day; (b) 7 days; (**c**) 14 days; (**d**) 28 days; (**e**) 60 days.

**Figure 15 materials-13-04238-f015:**
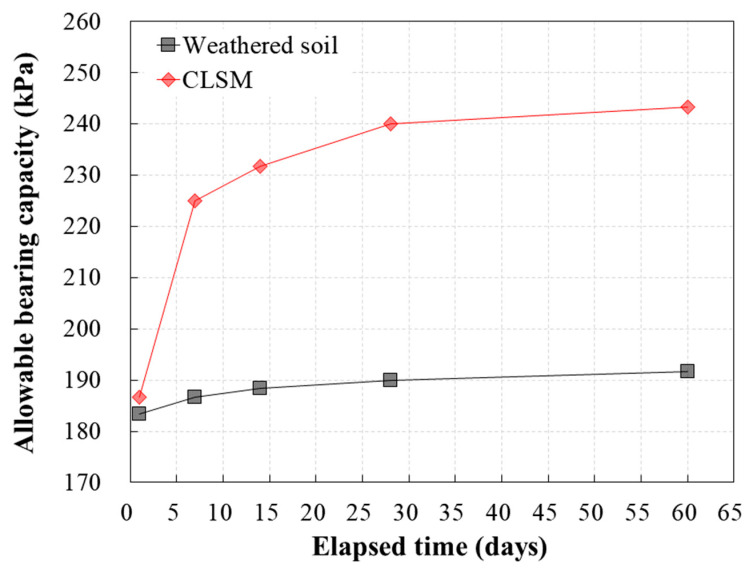
Comparison of allowable bearing capacity of weathered soil and CLSM.

**Table 1 materials-13-04238-t001:** Engineering properties of Jumunjin standard sand in Korea.

Specific Gravity (Gs)	Maximum Dry Unit Weight (γ_d(max)_; kN/m^3^)	Optimum Moisture Content (ω_opt_; %)	Internal Friction Angle (ϕ; °)	USCS *
2.65	16.9	9.4	33.4	SP **

* Unified soil classification system, ** Poorly graded sands and gravelly sands, little or no fines

**Table 2 materials-13-04238-t002:** Components of fly ash (FA).

Component	SiO_2_	Al_2_O_3_	Fe_2_O_3_	CaO	K_2_O	TiO_2_	MgO	Na_2_O	SO_3_	Loss on Ignition
FA (%)	75.94	14.70	3.85	1.47	1.11	0.83	0.6	0.54	0.46	0.5

**Table 3 materials-13-04238-t003:** Components of waste paper sludge ash (WPSA).

Component	SiO_2_	Al_2_O_3_	Fe_2_O_3_	CaO	K_2_O	TiO_2_	MgO	Na_2_O	SO_3_	Loss on Ignition
FA (%)	59.35	11.19	10.27	4.43	4.02	3.98	1.74	1.36	0.64	3.02

**Table 4 materials-13-04238-t004:** Mix designs.

Case Number	1	2	3	4	5	6	7	8	9	10	11	12
Cement (%)	5											
Sand (%)	35			40			45			50		
WPSA (%)	60	30	0	55	27.5	0	50	25	0	45	22.5	0
Fly ash (%)	0	30	60	0	27.5	55	0	25	50	0	22.5	45
**Case Number**	**13**	**14**	**15**	**16**	**17**	**18**	**19**	**20**	**21**	**22**	**23**	**24**
Cement (%)	10											
Sand (%)	35			40			45			50		
WPSA (%)	55	27.5	0	50	25	0	45	22.5	0	40	20	0
Fly ash (%)	0	27.5	55	0	25	50	0	22.5	45	0	20	40

**Table 5 materials-13-04238-t005:** Results of flowability tests.

Case Number	Mix Design				Water Content Required For Adequate Flowability (%)
	Cement (%)	Sand (%)	WPSA (%)	FA (%)	
01	5	35	60	0	32.0
02			30	30	29.0
03			0	60	27.0
04		40	55	0	31.5
05			27.5	27.5	28.5
06			0	55	26.0
07		45	50	0	29.5
08			25	25	26.0
09			0	50	25.0
10		50	45	0	29.0
11			22.5	22.5	26.0
12			0	45	25.0
13	10	35	55	0	31.5
14			27.5	27.5	28.5
15			0	55	26.5
16		40	50	0	30.0
17			25	25	27.5
18			0	50	25.5
19		45	45	0	28.5
20			22.5	22.5	25.0
21			0	45	24.0
22		50	40	0	28.0
23			20	20	25.0
24			0	40	24.0

**Table 6 materials-13-04238-t006:** Unconfined compressive strength test results.

Case Number	Unconfined Compressive Strength (MPa)				Case No.	Unconfined Compressive Strength (MPa)			
	1 Day	7 Days	28 Days	60 Days		1 Day	7 Days	28 Days	60 Days
01	0.12	0.59	1.00	1.10	13	0.13	0.76	1.70	2.11
02	0.16	0.60	1.30	1.31	14	0.21	1.10	1.50	2.50
03	0.56	0.86	1.50	1.53	15	1.40	3.60	4.50	5.00
04	0.08	0.31	0.53	0.76	16	0.17	0.87	1.15	1.30
05	0.05	0.50	1.00	1.20	17	0.16	0.91	1.84	1.91
06	0.50	0.81	0.90	0.91	18	1.30	3.90	4.79	5.20
07	0.09	0.49	1.10	1.20	19	0.18	1.00	1.37	1.20
08	0.13	0.52	1.20	1.22	20	0.30	1.20	2.47	2.50
09	0.65	0.90	1.30	1.29	21	1.20	3.50	5.84	5.96
10	0.06	0.30	0.54	0.70	22	0.32	0.79	1.00	1.20
11	0.04	0.50	0.80	0.72	23	0.44	0.92	1.82	1.88
12	0.40	0.60	0.80	0.73	24	1.10	3.00	4.81	5.06

**Table 7 materials-13-04238-t007:** Comparison of shear strength between weathered soil and CLSM with WPSA.

Classification		Normal Stress (kPa)			Strength Parameters	
		50	100	150	Cohesion (c; kPa)	Internal Friction Angle (ϕ; °)
Weathered soil	Shear stress (kPa)	57.8	77.6	120.1	22.9	31.9
CSLM after 7 days		90.0	129.0	162.1	54.9	35.8
CSLM after 28 days		144	196.7	225.5	107.2	39.2

**Table 8 materials-13-04238-t008:** Allowable bearing capacity of weathered soil and CLSM.

Classification		Elapsed Time (Days)				
		1	7	14	28	60
Allowable bearing capacity (kPa)	Weathered soil	183.3	186.7	188.3	190.0	191.7
	CSLM	186.7	225.0	231.7	240.0	243.3

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
