# Peer review of "Strength Characteristics of Controlled Low-Strength Materials with Waste Paper Sludge Ash (WPSA) for Prevention of Sewage Pipe Damage"

_materials, 2020, doi:10.3390/ma13194238_

Round 1

Reviewer 1 Report

Comments to the Authors

The presented article is a good research report. This paper reports interesting study about the controlled-low-strength materials (CLSM) with waste paper sludge ash (WPSA) and about it utilization as a sewage pipe backfill material. In the work have been used different methods to study the strength evaluation of CLSM with WPSA, flowability, compressive strength tests were conducted in accordance with standards.

The detailed suggestions are listed as follows:

1) in some sections (e.g. page 4 Table 1) standards are referenced without stating the code in the text. It would be better to state "standard sand according to ???? [??]

2) Table 2, Fig. 3-6 and others – need to add reference, or test methods. If so, please describe how this was done.

3) It would be appreciated if the authors tried to explain of the chosen compositions for example amount of WPSA and FA?

4) Table 6: Do you calculated change variation coefficient or standard deviation. If no, please explain it? I would round off to two digits and also present average and/or standard deviation.

5) Conclusions: You stated WPSA and FA which act as binders, had significant impacts on strength when the cement content was low Figure 4 and 6 show different mineral composition and intensities of peaks of phases. Why do you decided this?

6)  Please add information of technique description (producer, city, country). A style guide is included at the "For authors and editors".

Conclusion. The manuscript has scientific background and it will be attractive for readers, therefore I recommend accepting the paper after minor revision to make it more interesting.

Reviewer 2 Report

  • The article still needs some grammatical and syntax improvements. Use of English service center is recommended.
  • The abstract is written qualitatively. Majority of the qualitative statements should be modified for quantified result comparisons.
    • it was found that the allowable bearing capacity increased over time.
    • It was found that CLSM with WPSA performed better as a backfill and was more stable than soil immediately after construction
  • The introduction needs to be revised for higher quality language. Using separated paragraphs is encouraged, and it should be brief. In addition, the authors mentioned some works without stating about the contributions, pros and cons and the how the current work would address.
  • The authors mentioned about the status of the cavities in underground sewer systems. The environmental conditions have significant effects on the behavior of the concrete, which is suggested to be mentioned within this study. Following more recent works are recommend to be considered for this purpose.
    • Farzampour, A. (2019). Compressive behavior of concrete under environmental effects. IntechOpen.
    • Farzampour, A. (2017). Temperature and humidity effects on behavior of grouts. Advances in concrete construction, 5(6), 659.
  • In addition, recent works related to “improve the sewer concrete material” on is suggested to be considered as well.
    • Mansouri, I., Sadat Shahheidari, F., Hashemi, SMA., & Farzampour, A. (2020) Investigation of steel fiber effects on concrete abrasion resistance, Advances in concrete construction, 9(4), 367-374.
  • The concerting mixing design method should be defined with more details.
  • Figures should be more detailed.
  • Why the water content shown in figure 8 is significantly different between the cases
  • From the figure 9.f, why considerable improvement in compressive strength has occurred for the case 21 specimen?
  • Why the safety factor of 3 is used?
  • More quantitative conclusions for the figures 13,14, 15, and Table 8

Round 2

Reviewer 2 Report

Authors addressed the comments.